# The Role of Mass Media in Promoting the Dual Career of the Performance Athlete

**DOI:** 10.3390/bs13030196

**Published:** 2023-02-23

**Authors:** Ionut Onose, Raluca-Mihaela Onose, Beatrice-Aurelia Abalasei

**Affiliations:** 1Doctoral School of Sport Sciences and Physical Education, Faculty of Physical Education and Sports, “Alexandru Ioan Cuza” University of Iasi, 700554 Iasi, Romania; 2Faculty of Physical Education and Sports, “Alexandru Ioan Cuza” University of Iasi, 700554 Iasi, Romania

**Keywords:** dual career, mass media, student–athlete

## Abstract

Background: One of the most pressing problems for athletes is related to the end of their career, a situation that presents serious challenges in the attempt to enter the labor market in the absence of an adequate education. Despite the fact that successful sports students encounter some problems in terms of time allocated to school, consistent work will ultimately provide them with a rewarding career and, most importantly, a relevant social role to inspire other athletes who are at the beginning of their careers. A successful approach to dual career cannot take place if athletes and their entourage (coaches, parents, teachers) are not aware of its importance. By analyzing the concept from different angles by professionals (EU, governments, ministries), dual career can be implemented in an efficient way and much faster if the appropriate institutions are involved. The use of mass media to create interest and content in the sports field has managed to produce major interest among specialists in the field. Methods: In this study, the answers of 30 mass media specialists were analyzed to understand how “dual career” is promoted for the performance athlete in the mass media in Romania. In order to analyze if there is any connection between mass media, dual career and the profile of the performance athlete, we applied a chi-square test and Pearson and Spearman correlations. Results: The obtained results indicate a weak connection between the analyzed terms. Conclusions: The conclusions reached allow us to form an overview of the analyzed terms, managing to create directions for action in order to constantly improve the phenomenon in Romania.

## 1. Introduction

The transition to performance activity can be a characteristic action of every athlete and can be applied to any sports discipline as sports events are becoming more prevalent in modern society. For this reason, most of the time, young people are forced to agglomerate, or even overlap, their sports career with their education. Research [1] demonstrates that student–athletes need additional support in their academic career in order to cope with educational rigors. In this way, the beginning of a dual career for an athlete can take place. 

The term “dual career” refers (according to the European Guidelines for Dual Careers of Athletes) [2] to the attempt by athletes to combine sports performance with education or work and is a continuous source of concern among athletes [3]. Research shows the interest of specialists [4] in the dual careers of pupils/students. Regarding this topic, Vidal-Vilaplana et al. [5] analyze the upward trend in the specialized literature through numerous publications that present the dual career at the international level.

In this sense, in an analysis of the dual career, Gomez [6] states that a student–athlete represents: ”a person who is still in the field of education but also trains at the highest level” or ”a person who is a student or pupil and who participates in competitions or competes under the auspices of a sports federation, club or association.”

Crăciun is of the opinion that a true athlete: “makes it a priority to excel on a personal level, more than in the pursuit of victory, and focuses more on the journey than on the destination.” People involved in sports and physical activities are always interested to succeed [7].

Media play a shaping role in the development and sustainability of national culture, including dual careers [8]. In the last decades, there has been an increase in research in the field of the dual career of sports students; these scientific contributions have been identified and properly analyzed [9,10].

Sports and social media have developed in a perfect symbiosis as a result of technological development. Thus, the modern media landscape presents new sports and modes of interaction in the sports field [11,12]. With the advent of television, the relationship between sports and mass media is becoming closer and closer. One study [13,14] analyzed the way in which young athletes obtain their information about sports. Later, with the evolution of the sociology of sport, physical education, mass sport, etc., the relationship between sports and mass media has gradually become the object of academic research [15]. Clavio is of the opinion that in the last decades the amount of sports-related news has increased both in the traditional mass media and in new approaches to media, newspapers still maintaining an important role in the sports discourse and in society’s culture [16].

Since the first media platform appeared, the continuous development of technology has allowed people easy interaction, which has gradually replaced the traditional forms of communication. Snyder points out that the use of social networks is a concern for student–athletes, due to the fact that they are aware of their role in the social environment and the expectations that different sports organizations have regarding behavior on and off the field. In other words, at the global level, young people not only resort to social networks, but also use these platforms in their daily lives [17]. They can use the media not only as a tool for relaxation or knowledge, but also as a means to develop sponsorship opportunities or to promote brands [18]. Society’s passion and consumption pattern for sports, together with the technological advantages of the Internet, can promote behavior similar to public spheres, presenting useful information to users [19].

The role of mass media in sports is to communicate and transmit messages from sports organizations, or athletes, to the audience. Considering the effects of sports on different dimensions of personal and social life, the role of mass media as a mediator is essential [20]. How athletes are promoted, including the type of epithets used in their presentation or their actions on the field of play, is decisive in the formation of opinions of the individual [21].

Athletes are often present in the mass media through the prism of the results obtained. While great performance is constantly described through communication channels, also benefiting from the help of marketing and excessive advertising, from the point of view of their respective sports organizations [22], young athletes, at the beginning of their careers, can be easily drawn into the mirage of success expected. Since social networks can have a positive or negative impact on the development of athletes, Park et al. are of the opinion that understanding the phenomenon and optimal management of communication channels is key for employment in the next career [23].

Champions are balanced, resistant to stress, and not sensitive to pressure. They show increased attention and do not panic in difficult situations, their emotional reactions being appropriate to the stimuli, thus presenting a particularly well-defined psychological profile [24]. Bull [25] highlighted what exactly an athlete tries to fight for: pride, social recognition, and mastery. A profile of the performance athlete presupposes a value ideal, which must obviously be followed in the hopes of reaching or even surpassing it.

The situation in which world-class athletes (idolized by children and adults) present, with the help of social networks, their life and internal struggles, represents a moment of accessibility in their aura of invincibility, allowing fans, but also critics, to see and understand their future aspirations through the personal information posted [26].

The purpose of this study is to determine the role that mass media have/can have in the development of the dual career concept in Romania. In order to achieve this purpose, we surveyed 30 people who work in mass media regarding their opinion about dual career in Romania.

## 2. Materials and Methods

To carry out the research, which involves an exploratory study, qualitative analysis (categorical/thematic content analysis) required the completion of certain steps: identification of the problem in the specialized literature, with applicability in Romania; establishing the central question and the 2 associated sub-questions; selecting media specialists and obtaining their written consent (regarding participation in this study in accordance with the Helsinki Declaration); conception and application of the questionnaire; identification, based on the answers obtained, of the specifications necessary for this study; and interpretation of the obtained results (analysis of the interaction of the 2 concepts: mass media and dual career).

The answers highlight the specialists’ knowledge, presenting their vision of the categories used. Furthermore, being a qualitative analysis, it is subjective and conducted through the lens of the analyst’s ability to capture the key elements from the responses of those questioned.

In carrying out this study, we started from the following central question:

Do mass media play an important role in promoting “dual career” and the performance athlete’s profile?

The research had also 2 associated sub-questions:

Associated sub-question 1: Do mass media have the quality to promote “dual career”?

Associated sub-question 2: Is the promotion of the performance athlete’s profile determined by mass media?

In order to analyze a possible association between the independent variable (mass media) and the dependent variables (dual career and performance athlete profile), an interview guide was constructed, which contained 14 items.

The items were made so as to include all aspects of the phenomenon to be analyzed and are structured in the following categories: “dual career,” “performance athlete profile,” “mass media,” and “mass media tools” (Table 1). The questions addressed to the respondents were open and all participants were informed and asked for written consent before participation in this study. These categories are part of the verification strategy of the questions from which the qualitative study started, the conceptualization of the terms being adapted to the desire for analysis and synthesis of the phenomena used.

The subjects included in the research are 30 Romanian people (11 female and 19 male) who work in mass media (printed press—13 people, television—9 people, and radio—8 people), and they can be considered specialists in the field through the prism of the competencies they have and which guarantee their activity in the media field. Experience in the field is given by seniority in mass media: 16 of them have 0–10 years of experience, 5 have 11–20 years of experience, 6 have 21–30 years of experience, 2 have 31–40 years of experience seniority, and 1 has 41–50 years of experience.

## 3. Results and Discussions

Following the recording of the opinions formulated by the specialists, it is observed that most specifications are part of the mass media category (Table 2), because we are particularly interested in the role of the mass media in promoting dual career. In this sense, we made sure, through a larger number of questions, that we cover as many aspects of this field as possible.

### 3.1. For “Dual Career” Category

From the perspective of media specialists, the phrase “dual career” is present in society through an academic approach, which studies the concept itself. At the same time, public exposure is of overwhelming importance, constituting one of this study’s objectives.

Analyzing the answers obtained, we notice the fact that the largest number (24) was of people who associate dual career with a way to have simultaneous professions. At the same time, 23 times dual career is perceived as a combination of sports activity with another job, and only 7 references go to athletes who are also involved in the educational field (Figure 1).

The financial advantages that dual career implies are not an aspect to be neglected in the opinion of the respondents. Thus, it is found that the material part of the success obtained from sports is recognized, and possible gains that derive from the association of the sports career with an adjacent career are presented as a backup plan for the person’s future.

We find an uncertain approach regarding the phrase “dual career.” Although the term, in its essence, refers to duality, it is not clear, in the view of people who work in the media, in which direction it should be assimilated: we are talking about a person who has two jobs at the same time, who is a performance athlete but is also active in another field, or about an athlete who, concurrently with competitive life, has time for education.

### 3.2. For the “Performance Athlete Profile” Category

In the analysis of the type of athlete who falls into the “dual career” patterns, we encounter attributes that describe the ideal model of the athlete. The athlete must be adaptable to unforeseen situations, show impeccable conduct, and develop standard characteristics for which the ultimate goal is success in the form of constant performance.

Athletes who manage to be successful in a competitive activity and at the same time are aware of the importance of a broad approach to their training present a series of clearly defined characteristics in the answers obtained. The dominant trait, in the view of the respondents, is intelligence, followed by ambition and determination (Figure 2, attributes that are essential for a champion profile).

The attitudes and behaviors of the performance athlete can be found in the answers of those interviewed, which indicates that the mass media specialists correctly present the profile of the performance athlete, thus helping to educate and develop the vision of the winner in sports.

### 3.3. For “Mass Media” Category

The interview guide, being applied to professionals who carry out their activity in the media area, had questions that gravitated to their area of activity. For this reason, we obtained the largest number of specifications (210) in comparison with the specifications obtained in the other categories.

Through the lens of communication, the respondents’ vision regarding the role of the mass media in everyday life is outlined: the most important is the promotion of the aspects of utility in the studied concept (36 specifications, according to Figure 3), but also the informing of public opinion on the fields of interest and the activities carried out.

In building opinions about the performance athlete, the analysis of the interviews presents us with a clear situation: the athlete, along with the sports institutions that converge around them (clubs, ministries, managers, coaches, etc.), are the entities that can contribute to the development of the athlete image in society. In turn, these organizations, in order to be effective, need public visibility, and the aspects presented should be directed towards the positive values of the dual career.

A special situation is represented by the fact that in the answers obtained the specification "teaching staff" has a higher number than those obtained by "family" (parents). In this sense, the media specialists believe that the dual career approach should be constantly carried out by the athletes’ teachers.

Strategies regarding the support of performance sports can be carried out in two clear directions: the mass media discourse must be an adapted one, which highlights the needs, aspirations, and problems of the sports field, and it must act in the spirit of the positive values encountered in sports. It is easy to assume that these approaches do not increase the audience and the number of readers (views), but they represent an important step towards a state of affairs to be achieved in the interest of athletes.

### 3.4. For the “Media Tools” Category

Mass media can influence the population’s perception of a subject by reporting on the cause itself. Those who work in mass media have come to the conclusion that, despite the development of technology, social media have not yet completely captured the field (Figure 4).

The traditional media have, for the moment, their followers, who cannot be completely attracted to online platforms for retrieving information. That is why, in drawing up the conclusions of the preliminary study, substantial attention must be allocated to the traditional media category in order not to lose a large audience in the fight for informational supremacy.

Interpretation of the answers of the research subjects and their opinions regarding the profile of the performance athlete touches on aspects that we would not have expected. If the performances obtained can easily be considered the business card of an athlete, in addition to the skills that an athlete engaged in competitions must possess, it is surprising that, in the view of the respondents, the extra-sport activities should not be neglected. It is desired, in this way, to integrate and present the athlete with a touch of tangibility, thus leaving the aura of a hero that a champion usually creates.

Desirable behavior in sports elicited multiple and varied answers from the people interviewed. Dialogue with athletes is encouraged to obtain answers that can generate behaviors to follow, but presentation through personal example (of the athlete) can also represent a solution. Media promotion campaigns of the activity and the results obtained are also discussed, encouraging, in this way, the orientation towards a sports discipline.

Following the analysis of the obtained results (Figure 5), we can see that mass media are the optimal solution for presenting and raising awareness of dual career. The constitutive dimensions of the questionnaire must be built around it, through which we can capture the fundamental aspects of the problem: mass media must represent the fundamental element in the formation and promotion of the performance athlete’s profile.

To determine if there are associations between two variables, we will use a chi-square test. In other words, it will be analyzed if the independent variable “media” is associated with the dependent variables “dual career” and “performance athlete profile.” 

### 3.5. Associated Sub-Question 1 Testing

To answer the associated sub-question 1, the total specifications that were determined for the two variables, “mass media” and “dual career,” were used (Table 3).

Further in the analysis, the expected number of frequencies was calculated in order to achieve the association with the observed number of frequencies (Table 4 and Table 5).

We notice that for df = 1 (degrees of freedom), the value obtained is higher than the standard value for the significant threshold of 0.05 (3.841 and is taken from chi-square distribution), which means that there is an association between “mass media” and “dual career”. In order to check the relationship between the terms from another perspective, a Pearson and Spearman product-moment correlation was conducted to examine the relationship between mass media and dual career in the concept domain of physical education and sport.

Analyzing the obtained results, we can state that there is a slightly positive but insignificant correlation between the two concepts (Table 6).

Finally, comparing the data, we can say that between mass media and dual career there is a link, poorly represented statistically, which can indicate the direction of action in the social space so that these variables can be strongly correlated, so that they can, in the future, influence each other.

To answer the associated sub-question 2, the specifications for the media variables and the performance athlete’s profile were analyzed in order to determine if there is any association between them (Table 7).

The number of expected frequencies and the number of observed frequencies can be traced in Table 8 and Table 9.

We note in this case also the fact that for df = 1 (degrees of freedom) the result obtained is, for the significant threshold of 0.05, higher than the value in the chi-square distribution table, resulting in the fact that there is an association between “mass media” and “Performance athlete profile.”

The same Pearson and Spearman analysis was also carried out in the case of the relationship between the mass media and the profile of the performance athlete. The obtained results are presented in Table 10.

Analyzing the value of p, we can also state that there are slightly positive but insignificant correlations between the two notions.

The statistical values obtained also show us in this case the fact that there are connections between the mass media and the profile of the performance athlete in Romania, but they are poorly represented statistically.

The statistical analysis indicates that, in Romania, mass media, at this moment, do not have on their agenda the promotion of the dual career of the performance athlete. This is worrying, given the fact that young people consume mass media more and more under different forms (TV, social media, newspapers), and these awareness vectors are not aimed at presenting the advantages that derive from the dual career of athletes.

## 4. Conclusions

“Dual career” is a concept that is not publicized in Romania. Using the phrase on a large scale would allow athletes to understand why they need an alternative to sports activity. 

To be successful in this process, according to Condello et al. [27], all social actors must prioritize their political agenda in the implementation of programs that support dual career.

This alternative must be conceived during the period in which the athlete is active in the field of sports. In other words, media resources have a huge potential in the education of children and the sports culture of a nation [28]. Mass media, through their work tools, can achieve this desired effect, because the current society is continuously connected to information. Regardless of the communication channels to which we are anchored (social media, audio–visual, newspapers, etc.), the presence of the dual career can constantly be part of the champion’s consciousness.

Interest in sports encouraged the evolution of sports channels, and they even became vectors of influence of sports events [29]. Consequently, the influence of mass media on the concept of physical activity within society and on the individual and collective values that it claims is indisputable [30].

As a result of the research, several important conclusions emerge, which can be presented as decision-making factors in the educational and sports fields:“Dual career” should be intensively promoted on social media, because the target group of the research (athletes) predominantly uses related communication methods. In support of this statement, the interview respondents (mass media specialists), in proportion of 90%, recommended social media as a platform for awareness and presentation of the phrase “dual career.”Since a characteristic identified in athletes is “diversity,” this means that there is a basis for creating ways of expression in another field, which can provide them with an alternative career to sports. In this sense, their capacity for mobilization and ambition should not be underestimated; the sure thing they lack is the presentation of all the possibilities of achievement.An important role in the development of the “dual career” is played by sports institutions—clubs, associations, ministries, etc. It is necessary to go through all the stages of analysis and synthesis for the purpose of strong visibility, which uses and transmits the phrase (“dual career”) to all active athletes. Dual career, according to Robnik et al. [31], cannot be achieved without the help of parents and coaches, in a triangular relationship (athlete–parent, athlete–coach, coach–parent). At the same time, it is the duty of the teaching staff to intervene when they notice a syncope in the natural evolution of the pupils (students) and together with the subject’s family to discover and use the most effective means of understanding the road to success.

## Figures and Tables

**Figure 1 behavsci-13-00196-f001:**
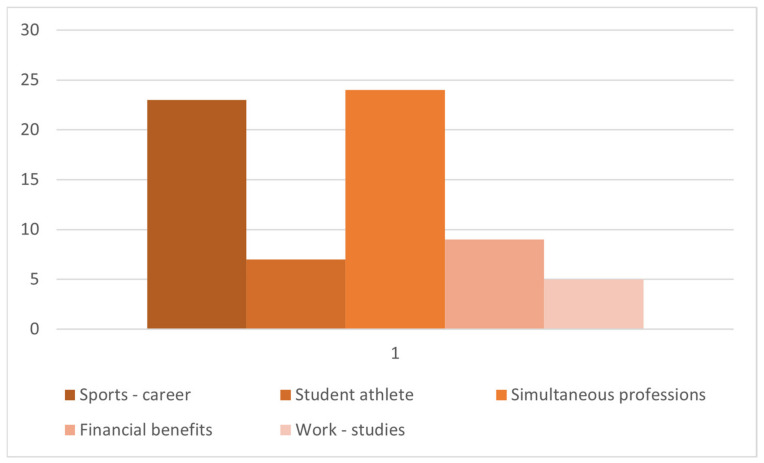
Distribution of specifications for “dual career” category.

**Figure 2 behavsci-13-00196-f002:**
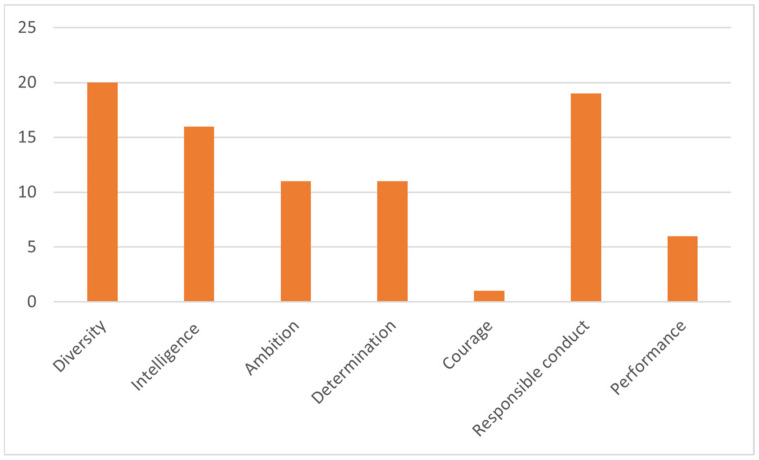
Distribution of specifications for the “profile of the athlete” category.

**Figure 3 behavsci-13-00196-f003:**
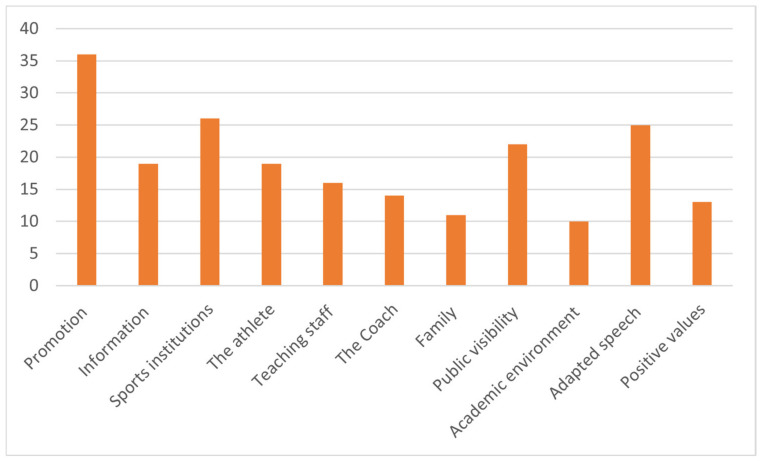
Distribution of specifications for “mass -media” category.

**Figure 4 behavsci-13-00196-f004:**
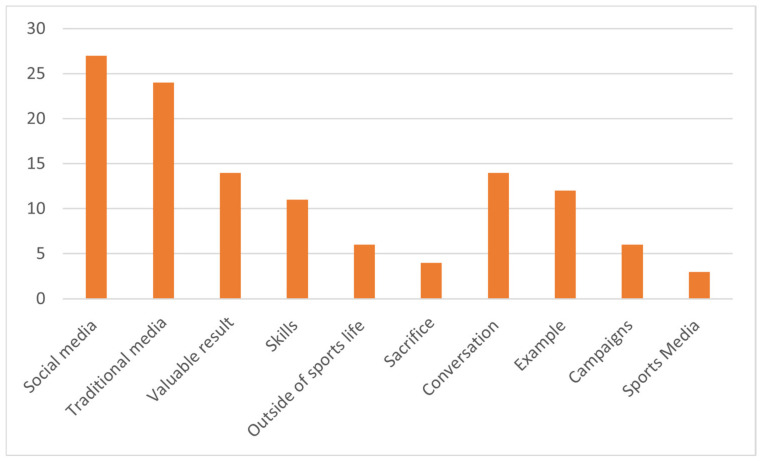
Distribution of specifications for the “media tools” category.

**Figure 5 behavsci-13-00196-f005:**
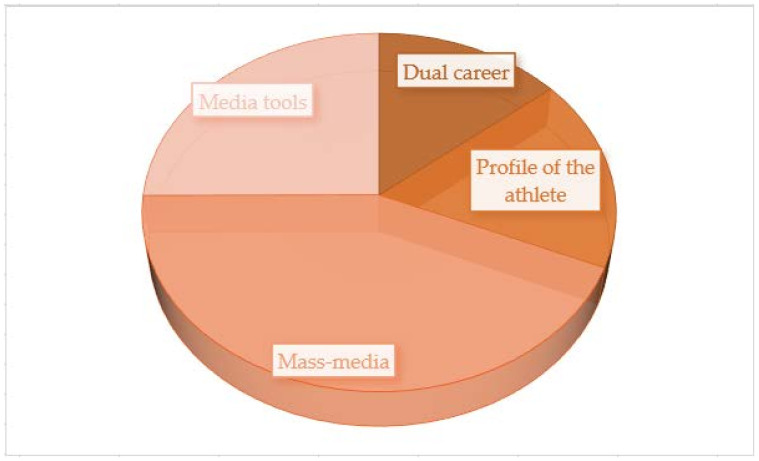
Hierarchy of content analysis categories for interviews.

**Table 1 behavsci-13-00196-t001:** Questionnaire items and categories.

Questionnaire Items	Categories
Item 1: “What do you understand by the phrase “dual career”? Item 2: “What are the advantages that the development of a “dual career” entails for an athlete”?	Dual career
Item 3: “How would you characterize athletes who are involved in the two dimensions: sport performance and academic career (when they are approached simultaneously)”? Item 4: “In your opinion, what type of athlete fits into the patterns of “dual career”? Item 5: “What are, according to you, the features of this athlete”?	Athlete profile
Item 6: “Where have you heard the notion of “dual career” used?” Item 7: “Who do you think has the main responsibility for the awareness of the development of a “dual career”? Item 8: “Who, in your opinion, is responsible for the proper promotion of the performance athlete”? Item 9: “What is the role of mass media”? Item 10: “What can you say about the way the mass media supports performance sports”? Item 11: “Can you create an optimal promotion solution for the performance athlete, which would represent a model for future generations”?	Mass media
Item 12: “What is the most effective channel for disseminating information in promoting the performance athlete (social media, print media, audio-visual)”? Item 13: “From the performance athlete’s profile, which aspects should these channels focus on”? Item 14: “What do you think are the basic tools of the mass media, which can generate desirable behaviors in sports (desired behaviors, to follow, etc.)”?	Mass media tools

**Table 2 behavsci-13-00196-t002:** Analysis of the distribution of themes and specifications at interview level.

Items	Category	Corresponding Themes	Specification
Item 1, Item 2.	Dual career (68)	1. Student–athlete—student (30)	Sports—career (23) Student–athlete (7)
		2. Vocational training (38)	Simultaneous professions (24) Financial benefits (9) Work—studies (5)
Item 3, Item 4, Item 5.	Performance athlete profile (84)	1. Qualities (59)	Diversity (20) Intelligence (16) Ambition (11) Determination (11) Courage (1)
		2. Psychological skills (25)	Responsible conduct (19) Performance (6)
Item 6, Item 7, Item 8, Item 9, Item 10, Item 11.	Mass media (210)	1. Communication channel (54)	Promotion (36) Information (19)
		2. Formation of opinions (86)	Sports institutions (26) The athlete (19) Teaching staff (16) The coach (14) Family (11)
		3. Dissemination (32)	Public visibility (22) Academic environment (10)
		4. Strategies (38)	Adapted speech (25) Positive values (13)
Item 12, Item 13, Item 14.	Media tools (121)	1. Keywords (51)	Social media (27) Traditional media (24)
		2. Performance athlete model/profile (35)	Valuable result (14) Skills (11) Outside of sports life (6) Sacrifice (4)
		3. Interviews (35)	Conversation (14) Example (12) Campaigns (6) Sports media (3)

**Table 3 behavsci-13-00196-t003:** Total number of specifications for “mass media” and “dual career.”

Category	Corresponding Themes	Specifications	Total
Media	Communication channel	54	210
	Formation of opinions	86	
	Dissemination	32	
	Strategy	38	
Dual career	Student–athlete	30	68
	Professional training	38	

**Table 4 behavsci-13-00196-t004:** Observed frequencies and expected frequencies for the variables “mass media” and “dual career.”

Categories	Observed	Expected	Difference
Media	210	139	71
Dual career	68	139	−71
Total	278		

**Table 5 behavsci-13-00196-t005:** Chi-square test value for the variables “mass media” and “dual career.”

Chi-Square	72.52
df	1

**Table 6 behavsci-13-00196-t006:** Pearson and Spearman correlation for dual career.

Correlations		Dual Career
Pearson r value	Mass media	0.1245
*p* value		0.5123
Spearman r value	Mass media	0.144
*p* value		0.4476

**Table 7 behavsci-13-00196-t007:** Total number of specifications for “mass media” and “performance athlete profile.”

Category	Corresponding Themes	Specifications	Total
Media	Communication channel	54	210
	Formation of opinions	86	
	Dissemination	32	
	Strategy	38	
Performance athlete profile	Qualities	59	84
	Psychological skills	25	

**Table 8 behavsci-13-00196-t008:** Observed frequencies and expected frequencies obtained for the variables “mass media” and “performance athlete profile.”

Categories	Observed	Expected	Difference
Mass media	210	147	63
Performance athlete profile	84	147	−63
Total	294		

**Table 9 behavsci-13-00196-t009:** Chi-square test value for the variables “mass media” and “performance athlete profile.”

Chi-Square	27
df	1

**Table 10 behavsci-13-00196-t010:** Pearson and Spearman correlation for performance athlete profile.

Correlations		Performance Athlete Profile
Pearson r value	Mass media	0.2584
*p* value		0.168
Spearman r value	Mass media	0.2449
*p* value		0.1922

## Data Availability

Data are unavailable due to privacy or ethical restrictions.

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
