# Peer review of "The Role of Mass Media in Promoting the Dual Career of the Performance Athlete"

_behavsci, 2023, doi:10.3390/bs13030196_

Round 1

Reviewer 1 Report

BehSci MS# 2182104: The Role of Mass-Media in Promoting the Dual Career of the Performance Athlete

In this manuscript the author(s) describe their study of the role that mass media plays for college athletes who are balancing their sport participation with their education.  This qualitative study included a sample of 30 professionals who worked in media fields.  After reading the manuscript I have some feedback that I hope will be useful.

1.     First, I would suggest that very early on the authors define what they mean by dual career.  This term is used to mean very different things in other scholarly contexts so might be confusing at first to the reader who understand that term differently.  Also, at times the term “dual career” is used, at others “double career.”  Is this intentional?  If so, please let eh reader know if you are trying to draw a distinction or if you are using them interchangeably.

2.     While I found the literature reviewed on pp. 1-2 to be interesting, I found myself reaching the Methods section not knowing what the purpose of the study was.  I suggest revising the literature review so that it leads logically and clearly to a stated research question.

3.     On page 2, lines 86-89 gendered pronouns are used.  I suggest revising so that this paragraph is not just referring to male identified people.

4.     I found myself confused at the presentation of the methods.  Generally, when using qualitative approaches, a research question or questions are posed but hypotheses are not generated.  I found it curious that they were offered here.  Next, Table 1 is presented as though you are conducting a quantitative study.  I’m not sure it is helpful here.  Finally, qualitative research includes important conventions around the researchers bracketing their experiences and describing/discussing their positionalities, so as to guard against reading their own biases and assumptions into the qualitative data.  It’s not clear to me that these steps were followed, which makes me wonder about the confidence we can have in the coding conducted and the results reported.

5.     Further, the steps taken to code the data need to be described in detail, including the description of the type of qualitative analyses conducted and a rationale for why you chose that analysis approach over others.

6.     Please provide information about your participants, including race/ethnicity data and a breakdown regarding in which media fields they work. Figures 1 and 2 are unnecessary and take up space.  Just describe this information in the text of the paper.

7.     The first statement in the results section mentioned that most of the “specifications” were in the mass-media category.  Interestingly, you asked 6 questions in the mass media category, but only 2-3 questions in every other category.  I would imagine that logically you would get more data in a category where you ask more questions. I’m not sure this is a meaningful conclusion.

8.     Also, it is unclear what is meant by “centralization of the opinions.”  Perhaps once the qualitative coding steps are detailed (see comment #5 above) this will become clear?

Reviewer 2 Report

Nice and well structured study on the role of mass-media in promoting the dual career of the performance athlete.

The manuscript contains original results and it is written in a very clear way.

The research design is appropriate and the methodology employed is adequately described.

It meets the criteria of scientific quality and relevance for this journal.

It is also suitably formatted for publication.

I recommend the manuscript for publication in the present form.

Reviewer 3 Report

Thank you very much for giving me the opportunity to read this article. 

The title of the paper is appropriate for the context and relevant for this research.

The theme is very actual and important and it is good that is discussed from the marketing specialist's point of views.

The aim is clear, but there are other statistical means that would help the authors to prove it. 

The list of references should be improved with publications from the last 3 years that discussed this subject during the pandemic period.

We will recommend rewriting the abstract to contain 4 main ideas (1) Background: Place the question addressed in a broad context and highlight the purpose of the study; (2) Methods: briefly describe the main methods or treatments applied; (3) Results: summarize the article’s main findings; (4) Conclusions: indicate the main conclusions or interpretations.

The introduction evaluates very well the state of the art, emphasizing the role of mass media in influencing dual careers for athletes.  

The authors do not meet the requirements of the methodology. They should talk about sample representativeness, talk about the survey questions (open/closed.... types of answers). Did the authors obtain the ethical committee agreement and respect GDPR?

The study methods are valid and reliable and can be replicated, but insufficient. The authors can implement correlation analysis, factorial analyses, regression, etc.

The authors should present the p-value of ChiSquare inferential analysis.

In the methodology section tables and figures are relevant and clearly presented. Titles, columns, and rows labeled are correctly and clearly presented

The authors tried to interpret the data. The text in the results is not repetitive.

The results are discussed. A new perspective and multiple angles might be placed into context without being overinterpreted. 

Mainly the conclusions supported by references or results.

The study design was appropriate to answer the aim.

Congratulations to the authors.

Round 2

Reviewer 1 Report

Thank you for addressing the comments from the first review.  I found the manuscript much improved.  I would suggest, however, more details be provided about the steps taken in the qualitative analysis, including whether the authors engaged in bracketing prior to analysis.  

Author Response

Dear Professor,

Please find attached our answers to your suggestions. We   would   like   to   thank   you   for   the constructive feedback on the previous version of our manuscript.

  1. Thank you for addressing the comments from the first review. I found the manuscript much improved.  I would suggest, however, more details be provided about the steps taken in the qualitative analysis, including whether the authors engaged in bracketing prior to analysis. 

Thank you for the suggestions, we added the steps followed in carrying out the research. You can find them from line 111-121.

Reviewer 3 Report

Congratulation!

Author Response

Dear Professor,

We   would   like   to   thank   you   for   the constructive feedback on the previous version of our manuscript.

Thank you again for your patience, help and for the kind words!

The authors
